# Phytochemical Profiling, and Antioxidant Potentials of South African and Nigerian *Loranthus micranthus* Linn.: The African Mistletoe Exposé

**DOI:** 10.3390/plants12102016

**Published:** 2023-05-18

**Authors:** Siyabonga Hlophe, Kokoette Bassey

**Affiliations:** Department of Pharmaceutical Sciences, School of Pharmacy, Sefako Makgatho Health Sciences University, P.O. Box 218, Pretoria 0208, South Africa; siya4488@gmail.com

**Keywords:** *Loranthus micranthus* Linn., phytochemical profiling, antioxidant, interchangeable use

## Abstract

*Loranthus micranthus* Linn. (Loranthaceae) is a botanically significant hemiparasite that grows on tree branches or trunks and is used in traditional medicine. This study compares the antioxidant activity and qualitative phytochemical screening of Nigerian and South Africa *Loranthus micranthus* Linn. Standard techniques for phytochemical screening were deployed while thin layer chromatography (TLC) bio-autography was utilized to analyze antioxidants qualitatively. Quantitative antioxidant analysis was performed using 2,2-diphenyl-1-picrylhydrazyl (DPPH); hydrogen peroxide (H_2_O_2_) free-radical scavenging; and ferric chloride reducing power. The results of qualitative phytochemical screening revealed the presence of flavonoids; glycosides; saponins; phenolic compounds; phlobatannins; tannins; and terpenoids. As for the antioxidant potentials of the four extracts—i.e., Nigerian dichloromethane (NGDCM); South African dichloromethane (SADCM); Nigerian methanol (NGMeOH); and South African methanol (SAMeOH)—the SADCM extract had more creamy bands compared to the Nigerian, thus indicating potentially more antioxidant compounds. A better complementary antioxidant potential was observed for the Nigerian methanol extracts over their South African counterparts. The DPPH quantitative analyses underpin that the SADCM exhibited greater scavenging activity compared to the NGDCM, but this was less than the gallic acid control, with the highest activity at a concentration of 0.2 mg/mL and 0.4 mg/mL, respectively. However, as the concentration increased from 0.6 to 1.0 mg/mL, the SADCM again dominated in its antioxidant potential over all the analytes. The half maximal inhibitory concentration (IC_50_) values obtained were [SADCM = 0.31 mg/mL, NGDCM = 0.51 mg/mL, SAMeOH = 0.51 mg/mL, NGMeO = 0.17, gallic acid = 1.17 mg/mL, and BHT = 1.47 Mg/mL)]. Both the H_2_O_2_ scavenging and the Fe^3+^ to Fe^2+^ reduction assays mirrored a similar trend in the antioxidant potentials of all the analytes except for the BHT, which performed better in the ferric chloride reduction assay at a concentration of 0.2–06 mg/mL. Based on the facts gathered, it can be inferred that the South African and Nigerian *Loranthus micranthus* Linn. are chemically equivalent. This is in support of their similar morphology and taxonomical classification, notwithstanding the environmental, biological, and edaphic impacts experienced by each plant.

## 1. Introduction

Plants have historically been essential as sources of new compounds, with the potential to be funneled into drug pipelines for the creation of safe, efficacious, and cost-effective medicines over the past two decades [1]. The early ethnopharmacological investigations in Sub-Saharan Africa found over 100 plant species having antihypertensive action in animals and humans [2]. One of these plants is *Loranthus micranthus* Linn., also known as mistletoe tree, of the Loranthaceae family. The plant is a semi-parasitic shrub that obtains nutrients and support from a variety of trees, including *Kola acuminata* and *Persia* sp. [3]. For many years, both Nigeria and South Africa have documented the plants as high-quality traditional medicines in the treatment of diabetes, hypertension, hypolipidemia, oxidative stress related diseases, diarrhea, schizophrenia, and as an immune booster [4,5,6]. It has been discovered in Nigeria and other African countries that the use of the aqueous extracts of mistletoe may rid the body of the causes of hypertension, diabetes, and other metabolic illnesses [4], as well as help in circumstances when immunomodulation is required [4]. However, the consistent consumption of mistletoe plant and berry could result in seizures, bradycardia, abnormally high and low blood pressure, and vomiting [7]. This is partly due to the fact that the chemistry of the plant has not been fully explored.

A thorough literature search proves that the Nigerian *Loranthus micranthus* Linn. has been better studied than its South African counterpart. For instance, only a folkloric use comparison study has been done for both species [4,6]. Furthermore, the biological activities of the Nigerian *Loranthus micranthus* Linn., as summarized by Moghadamtousi and co-workers [6], include anti-diabetic, antimicrobial, immunomodulatory, antihypertensive, antioxidant, and antidiarrheal activities, just to mention but a few. In addition, more work has been reported regarding the isolation, purification, and elucidation of the phytoconstituents that are purportedly responsible for the biological activities of Nigerian *Loranthus micranthus* Linn., unlike that of the South African ecotype. Examples of isolated compounds are 3, 4, 5-trimethoxybenzoate, lupeol, 7α, 15β-dihydroxy lupeol palmitate, friedelin, and 3-methoxy quercetin [8,9].

Despite this loophole, several commercially available products formulated with mistletoe extracts, in addition to those traditionally concocted, are widely consumed [10]. It is a recorded fact that the South African and Nigerian markets are the two largest and account for the most vibrant economic hubs in Africa [11]. As a results of their rivalry in sales and profit making, one may envisage that raw materials and products travel be- tween the two countries. Furthermore, the *Loranthus micranthus* Linn. plants from these two countries are taxonomically classified as African mistletoe, without a basic comparison of their chemistry. Very serious and dire consequences from the research negligence could result in product adulteration, and even death from the use of such products. An attempt to redress some of the underlying problems and improve on the safety, quality, and commercialization of mistletoe-based products was the rationale for this study.

## 2. Results

### 2.1. Qualitative Phytochemical Screening

Phytochemicals present in *Loranthus micranthus* Linn. leaves are composed of flavonoids, tannins, saponins, terpenoids, and cardiac glycosides (Table 1), according to our qualitative phytochemical screening. It has been reported that alkaloids are efficacious as medicines for a variety of illnesses because they possess anti-inflammatory, antibacterial, and anticancer properties, and may also serve as a good antidote in suppressing cancer progression [12]. In the present study, negative outcomes were observed for the presence of alkaloids in the NGDCM, NGMeOH, SADCM, and SAMeOH mistletoe tree extracts investigated. This comes as a surprise as the literature information from *Loranthus micranthus* Linn. studies reported the presence of alkaloids in the DCM and MeOH extracts [13]. Perhaps their presence in the samples we analyzed was below the limits of detection by TLC, which is a low-resolution technique of analysis. Flavonoids exhibit biological properties that help in the prevention and treatment of a variety of diseases, including cancer, oxidative-stress-related disease, and inflammation, it occurs in moderate quantity in the MeOH extracts from both countries. Both South African and Nigerian extracts revealed richness of cardiac glycosides, whereas Nigerian DCM and MeOH extracts showed greater richness compared to the SADCM and MeOH extracts. This observation partly validates the use of the extracts against cancer, as it is known that cardiac glycosides have potent activity against every cancer cell line [14,15].

Saponins are strongly present in the methanol extract compared to the DCM extracts. The amounts of these phytochemicals further support the use of mistletoe extracts in alleviating cancer because saponins have been shown to reduce the risk of cancer, lower blood cholesterol levels, and lower blood glucose levels [16]. In addition, saponins have also been shown to have anti-inflammatory, hypocholesterolemia, and immune-stimulatory properties [17].

A similar observation was made about the presence of phenolic compounds in the analyzed extracts. Whereas both SAMeOH and NGMeOH extracts revealed a stronger presence of phenolic compounds, only trace amounts were detected in the DCM extracts. As for phlobatannins, the Nigerian and South African DCM extracts indicated no presence of phlobatannins, while the methanol extracts from both countries contain a moderate amount of phlobatannins. Again, the positive test for phlobatannins supports the use of mistletoe extracts in the traditional management of wound healing as an anti-inflammatory and analgesic agent [18]. The test for tannins showed excellent results, but tannins were present more in the polar MeOH extracts than in the non-polar DCM extracts. Given that tannins are known to have biological properties that aid in the prevention and treatment of a variety of disorders and have been found to lower total cholesterol, lower blood pressure, and stimulate the immune system [19], their strong presence in the mistletoe extracts underscores the clinical importance of the plant.

Regarding the test for terpenoids, South African and Nigerian DCM extracts indicated trace amounts, while the MeOH extracts indicated a moderate quantity of these phytochemicals. This is potentially a positive outcome for the plant because terpenes usually exhibit antibacterial properties against both antibiotic-susceptible and antibiotic-resistant bacteria, mostly through their ability to stimulate cell rupture and impede protein and DNA synthesis [20]. Our findings are consistent with those previously reported [21,22]. Even though it is well documented that anthraquinones exhibit anti-inflammatory, antioxidant, antibacterial, antiviral, anti-osteoporosis, and anti-tumor activities [23], they were not detected in all the extracts analyzed. Except for anthraquinones, all the phytochemicals determined from this study were reported from *Loranthus miscanthus* (L.f) Ettingsh from India [24].

### 2.2. Qualitative and Quantitative Antioxidant Analysis of Nigerian and South African Mistletoe Extracts

#### 2.2.1. Qualitative DPPH Radical Scavenging Activity of Nigerian DCM Extract

DPPH can accept an electron or hydrogen radical to form a stable diamagnetic molecule. Changes in color, from purple to yellow, indicate a decrease in absorbance of a DPPH radical. This demonstrates that the antioxidant agents found in a plant extract mixture interact with the hydrogen radicals [25]. The results for the NGDCM extract are displayed in Figure 1A. The cream coloration against the purple background on the TLC plate is a positive test for the presence of antioxidants in the mistletoe extract. A TLC chromatogram of South African mistletoe DCM extract (Figure 1B) was also evaluated for qualitative evidence of antioxidants by spraying the fourth plate with 0.2 mM DPPH solution. Creamy bands indicated the presence of antioxidants [26] at different Rf values. These antioxidants were thought to be phytochemicals, including carotenoids such as astaxanthin and lycopene, based on the red bands when the plates were viewed at 366 nm. The blue bands at Rf of 0.78 suggest the presence of antioxidant phytochemicals, including polyphenolic compounds such as flavonoids, procyanidins (monomeric and oligomeric form), flavonols (i.e., kaempferol, quercetin, and myricetin), and phenolic acids [27].

Although the NGDCM and SADCM extracts indicated the presence of antioxidant compounds, there were variations in these compounds from both extracts. As evident in Figure 1, the SADCM has more antioxidant compounds than the Nigerian extracts. These variations are summarized in Table 2 using the Rf values of the respective bands. However, variations in the group of phytochemicals were very minor, as indicated by a *p* value of 0.06 calculated from the Rf values of the antioxidant bands.

#### 2.2.2. Qualitative DPPH Radical Scavenging Activity of Nigerian and South African Methanol Mistletoe Extract

A TLC chromatogram of NGMeOH and SAMeOH mistletoe extract was qualitatively analyzed for evaluation of antioxidants. The creamy bands on the Nigerian extract at Rf values of 0–0.1, 0.15, 0.2, 0.31, and 0.43 were visible. One can see that the cream color on the NGMeOH tends to fade as the Rf value increases, or rather as the compounds’ polarity decreases, thus suggesting that more polar compounds indicated stronger antioxidant potential.

Compared to the SAMeOH, the creamy bands were not visible, even though many bands were visible when the plate was viewed at 366 nm. This could imply that the compounds in SAMeOH extracts are polar but not necessarily having a strong ability to scavenge free radicals, or that the compounds could not react with DPPH because they are present in meager quantities in the extract. The variations in the different phytochemicals in both the NGMeOH and SAMeOH extracts and the bands with antioxidant—or otherwise—potentials are summarized in Table 3 below. The observed variations were very significant, with a value of *p* = 0.002.

### 2.3. Quantitative Analysis of Nigerian and South African Mistletoe Extracts

#### 2.3.1. Quantitative Analysis of SADCM and NGDCM Extracts

The free radical scavenging activity of NGDCM and SADCM extracts, gallic acid, and butylated hydroxytoluene (BHT) as standards were measured spectrophotometrically at 517 nm (Figure 2). The DPPH radical scavenging percentage inhibition of the SADCM mistletoe extract gave evidence in increments of the concentration, showing dominance against the NGDCM mistletoe, although, at a concentration of 0.4 mg/mL, the NGDCM presented a slight increase in antioxidant activity compared to SADCM. However, the gallic acid standard demonstrated the highest activity, at a concentration of 0.2 mg/mL and 0.4 mg/mL, but as the concentration increased from 0.6 to 1.0 mg/mL, the SADCM showed further dominance in its antioxidant potential. The lowest activity was demonstrated by the NGDCM extract, regardless of the increase in concentration. In terms of the percentage (%) inhibition, the SADCM extract exhibited greater percentage inhibition of DPPH compared to the NGDCM at a concentration of at 0.4 mg/mL.

#### 2.3.2. Quantitative Analyses of Nigerian and South African Mistletoe Methanol Extracts

The DPPH radical scavenging percentage inhibition of SAMeOH and NGMeOH extract compared to two standards, namely gallic acid and BHT (butylated hydroxytoluene), was also measured at 517 nm spectrophotometrically. The NGMeOH extract showed dominance in the DPPH radical scavenging percentage inhibition at a concentration range of 0.4–1.0 mg/mL, with a percentage inhibition rate of 50–90% (Figure 3). This was followed by the SAMeOH at the same concentration range but with a percentage inhibition of 30–88%, further shedding light on the false negative results obtained from TLC analysis of the extract. 

#### 2.3.3. Hydrogen Radical Scavenging Activity of NGDCM and SADCM Extracts

In Figure 4, the hydrogen peroxide free radical scavenging activity was performed, where the gallic acid and BHT are standards compared against the NGDCM and SADCM mistletoe extracts. The SADCM showed a concentration (0.4–1.0 mg/mL)-dependent inhibition of 60–87 % at 560 nm compared to the NGDCM and standards. This does not come as a surprise, as evidence from the qualitative study gave a similar outcome: at a concentration of 0.2 mg/mL, the NGDCM and SADCM extracts displayed similar hydrogen scavenging power, where SADCM had 18.2% compared to the NGDCM with 16.94%. The NGDCM showed the second-highest percentage dominance in scavenging ability, as compared to the standards and SADCM.

#### 2.3.4. Hydrogen Radical Scavenging Activity of Nigerian and SA Methanol Extracts

The ability of SAMeOH and NGMeOH extracts to scavenge hydrogen peroxide was determined according to the method described by Mapfumari and co-workers [13]. Again, in agreement with the other assay methods, the NGMeOH exhibited high activity throughout compared to the SAMeOH sample. 

As displayed in Figure 5, at a concentration of 0.2 mg/mL, the NGMeOH, at 84%, performed slightly better than the South African sample with an inhibition of about 78%. A similar observation is seen at a concentration of 0.4 mg/mL. However, both samples had an equal inhibitory effect at 0.6 mg/mL. Interestingly, the extracts performed exceptionally better than the standard antioxidant compounds. Traditional decoctions of mistletoe are usually prepared using a consumable alcohol such as ethanol, whose polarity is like that of the methanol used in this study. Consumers of such traditional decoctions will benefit from the antioxidant agents in the extracts.

#### 2.3.5. Ferric Chloride Reducing Power Assay

Unlike the DPPH and H_2_O_2_ free radical scavenging assays, which usually characterize good antioxidants as having high percentage inhibition values, good antioxidant agents in the ferric chloride reduction assay are those components with the lowest percentage reduction values. In this study, the SADCM and NGDCM extracts and the gallic acid standard tend to perform equally at a concentration range of 0–0.8 mg/mL with a reduction power of 39–49% (Figure 6). While the BHT standard exhibited the best antioxidant potential in this assay, this observation contravenes the trend observed with the DPPH and H_2_O_2_ assay, where antioxidant activity was from the standards to NGDCM and then to SADCM.

A similar slight deviation in the trend in the antioxidant potentials of NGMeO, SAMeOH, and standards was also observed in the reduction of the Fe^3+^ to Fe^2+^ assay (Figure 7). The NGMeOH, SAMeOH, and gallic acid again displayed similar reducing power at 0.1–0.6 mg/mL, except for SAMeOH at 0.4–0.6 mg/mL. Like the result obtained for the DCM extracts, the BHT equally exhibited the best antioxidant potential, with a reduction power of 20% at 0.2 mg/mL. The marginal shift in the trend in the antioxidant potentials of mistletoe extracts from the DPPH, H_2_O_2,_ and ferric reducing power was previously reported [13], and the redox reaction mechanism was proposed as the reason for it. The potency of inhibition by the extract (IC_50_) values was determined and the results obtained support the pattern in the antioxidant potentials of the four extracts and standards, as summarized in Table 4.

## 3. Discussion

The phytochemicals of *Loranthus micranthus* Linn. leaves are composed of flavonoids, tannins, saponins, terpenoids, and cardiac glycosides, according to a qualitative study conducted previously [28]. In the current study, there was no positive test for alkaloids for both the Nigerian and South African mistletoe tree extracts, which contradicts many studies conducted before this one [13,25]. This observation may partly be a result of the alkaloids being possibly below the limit of detection for the analysed extracts. On the other hand, the tannins showed excellent results, where all four extracts showed the same strength or availability of tannins, as previously reported [29]. For the phlobatannins, both Nigerian and South African DCM extracts showed no trace of phlobatannins, whereas methanol extracts for both countries showed the presence of phlobatannins. For the flavonoids, the experiment showed strong availability of flavonoids in both Nigerian and South African methanol extracts, in agreement with [30], and lesser presence compared to the DCM extract. Saponins show greater presence for the South African DCM and MeOH compared to the Nigerian DCM and MeOH extracts. On the other hand, there was no significant difference between the SA and Nigerian phytochemicals with respect to terpenoids and cardiac glycosides, as they both equally exhibited a strong presence of phytochemicals. The effect of antioxidants on DPPH is thought to be due to their hydrogen donating ability [31]. Radical scavenging activities are very important in preventing the deleterious role of free radicals in different diseases, including cancer. Based on a DPPH radical scavenging activity comparison of the SADCM and NGDCM, the SADCM showed stronger scavenging activity than NGDCM; on the other hand, the NGMeOH showed dominant scavenging potential over the SAMeOH. The results obtained in this study suggest that all the extracts from SA and Nigerian mistletoe trees showed radical scavenging activity through their electron transfer or hydrogen donating ability. For the hydrogen radical scavenging activity, the SADCM extract showed strong scavenging ability compared to the NGDCM; on the other hand, the NGMeOH exhibited strong hydroxyl scavenging activities. The results of this study support the use of both Nigerian and South African mistletoe trees as easily accessible sources of natural antioxidants, for use as possible food supplements or in the pharmaceutical industry. As for the assay, the South African extract performed better, but not significantly, than its Nigerian counterpart.

Mistletoe’s use in the traditional management of hypertension, diabetes, and schizophrenia and as an immune system booster has been well reviewed [32]. However, information about the phytochemicals that elicit the myriad of health benefits from the plant is mainly documented for the Nigerian species. Some of the isolated and elucidated compounds of the Nigerian mistletoe were well listed—including polyphenols, polyphenol glycosides, terpenoid esters, steroids, triterpenoids, steroids, and alkaloids—in the same review. The identification of these compounds in the South African *L. micrathus* Linn. *via* LC/GC-MS analysis, in terms of their isolation, purification, and structural elucidations, is underway in our laboratory.

## 4. Materials and Methods

### 4.1. Plant Collection, Preparation, and Storage

A mistletoe plant was identified by a botanist in the Department of Pharmacognosy, Faculty of Pharmacy, Obafemi Awolowo University, Ile-Ife, Osun State Nigeria (7°31′14.7612″ N,4°31′49.1340″ E). The plant was collected, sorted, processed, dried, and powdered at the Collect and Curate Initiative Centre of the Pharmacognosy Department. A total of 2.0 Kg of the plant powder was purchased on 1 March 2019 and the sample voucher NGMTT01 specimen was allocated. The leaf samples from South Africa, on the other hand, were collected from the Orchards residential area, in the northern part of Pretoria in the Gauteng Province (25°38′57.9″ S 28°05′32.6″ E), in autumn using a convenience sampling method. An indigenous knowledge system (IKS) practitioner (Traditional healer) and the South African Biodiversity Institute (SANBI) identified the specimen, and SAMTT01 was assigned to the sample. Both voucher specimens were deposited in the Herbarium of the Pharmaceutical Sciences Department of the School of Pharmacy, Sefako Makgatho Health Sciences University. The plant sample was then dried at room temperature, grinded to powder using a Polymix Laboratory Dry Mill Drive Unit (Polymix™ PX-MFC 90 D, Kinematica AG, Luzern, Switzerland) at 3500 revolutions per minute (RPM) and stored in brown paper bags in the dark for further use.

### 4.2. Extraction of the Plant Material and TLC Analysis of the Extracts

A total of 100 g of the powdered plant material was placed in 100 mL of dichloromethane in a 250 mL Erlenmeyer flask and placed in an ultrasonic bath set at the boiling point of the extracting solvent. The extraction was performed for 30 min. Thereafter, the mixture was filtered using Whatman number 4 filter paper. The filtrate was then stored, and the extraction process repeated 2 more times. The combined filtrate was evaporated to dryness using a rotary evaporator to generate the dry mass of the dichloromethane extract. A separate 100 g of powder plant was extracted with 100 mL of ethanol using the same protocol. The percentage yield of the dry dichloromethane and ethanol extract was then calculated.

Following the extraction procedure, 1.0 mg of the extract was dissolved in 5 mL of DCM for the DCM extract and 1.0 mg of the methanol extract was dissolved in 5 mL of methanol. The solution was vortexed and filtered. The extract solution was then spotted on the 352 TLC plate and, after drying for 5 min, the dried TLC plate of the methanol extract was developed using ethyl acetate: hexane (1:3 *v*/*v*) as the mobile phase and the DCM extract was developed using hexane: acetone: ethyl acetate (8:1:1 *v*/*v*/*v*) as the mobile phase. After developing the TLC plate in their respective mobile phases, the chromatograms were viewed initially without chemical treatment, at a short wavelength of 254 nm and a long wavelength of 366 nm light. For the spraying procedure, the developed, dried TLC plate was positioned on a sheet of filter paper beneath a fume. The sprayer was filled with 0.2 mg/mL of DPPH solution and sprayed from around 15 cm using a rubber ball/pump.

### 4.3. Qualitative Phytochemical Screening

The qualitative phytochemical screening was performed using standard protocols, as described in Table 5.

### 4.4. Qualitative DPPH Free-Radical Scavenging Activity Assay of the Mistletoe Extracts

This assay was conducted by weighing 10.0 mg of each extract of SA (DCM and MeOH) and Nigerian (DCM and MeOH). Each extract was dissolved in 10.0 mL of either DCM or MeOH, vortexed, and filtered before use. The TLC plates were prepared by drawing a 1 cm line from the bottom of the aluminum pre-coated TLC plates after the extract was spotted on the TLC at the drawn baseline and was allowed to develop with the use of a different solvent system for the different extracts. The dichloromethane extract was developed with hexane: Acetone: ethyl acetate (8:1:1 *v*/*v*/*v*) and the methanol extracts was developed with hexane: ethyl acetate (3:1 *v*/*v*). The developed plates were each then observed under normal light for visible bands and then visualized under an ultraviolet lamp at 254 nm and 366 nm. The plates were thereafter sprayed with 0.4 mM solution of 2,2-diphenyl-1-picrylhydrazyl (DPPH) dissolved in methanol. The formation of a yellow-to-cream spot(s) against the purple background was taken as a positive indication for the presence of antioxidant compounds.

### 4.5. Quantitative Phytochemical Screening

#### 4.5.1. DPPH Free Radical Scavenging Activity of the Mistletoe Extracts

This was achieved following a method described in [14]. A variety of concentrations ranging from 0.2 mg/mL to 1.0 mg/mL were prepared for both the non-polar dichloro- methane and the polar methanol extracts. Additionally, a DPPH solution was created with a concentration of 0.2 mg/mL. To test the extracts’ antioxidant activity, 1.0 mL of the DPPH solution was mixed with 1.0 mL of the extract solution in a test tube, and the contents were thoroughly mixed and vortexed before being placed in a dark cardboard for 30 min. The spectrophotometric absorbance of the different concentrations was measured at 517 nm using a 96-well microplate-reader spectrophotometer (SprectraMax^®^, Molecular Devices, CA, USA). Gallic acid, diosgenin, and butylated hydroxyl toluene (BHT) were used as reference standards at the same concentration. The percentage of radical scavenging activity of the extracts was calculated using Equation (1) below:%DPPH radical scavenging activity = A_0_ − A_s_/A_0_ × 100(1)
where A_0_ is the absorbance of the negative control and As is the absorbance of the extracts/standards.

#### 4.5.2. Hydrogen Radical Scavenging Activity

The hydrogen peroxide scavenging potential of the mistletoe tree extracts was assessed using the method described in [14]. To conduct an experiment, a 2 mL solution of hydrogen peroxide (20 mM) was prepared in a phosphate buffer saline with a pH of 7.40. To this solution, varying concentrations (ranging from 0.2 mg/mL to 1.0 mg/mL) of extracts from stock solutions were added in increments of 1.0 mL. The resulting mixture was mixed thoroughly using a vortex and incubated for 10 min before measuring the absorbance at 560 nm using a spectrophotometer. To ensure accuracy, the reference standards used for this experiment were 1.0 mg/mL of both gallic acid and butylated hydroxyl toluene (BHT). The percentage of hydroxyl radical scavenging activity was calculated according to the following formula:% Hydroxyl radical scavenging activity = A_0_ − As/A_0_ × 100(2)
where A_0_ is the absorbance of the negative control and As is the absorbance of the extracts/standards.

#### 4.5.3. Ferric Chloride Reducing Power Assay

The ferric chloride reducing power assay of samples was evaluated using the method in [14]. To start the experiment, the various extracts were dissolved in their respective solvents. Then, a range of concentrations from 0.2 mg/mL to 1.0 mg/mL was prepared. Each concentration was mixed in a test tube with 2.5 mL of 0.2 M phosphate buffer (pH 6.6) and 2.5 mL of 1% (*w*/*v*) potassium ferricyanide (K_3_Fe (CN)6). The contents were mixed and incubated at 50 °C for 20 min. After this, 2.5 mL of 10% (*w*/*v*) trichloroacetic acid was added and the mixture was centrifuged for 10 min at 3000 rpm. The upper layer of the resulting solution (2.5 mL) was mixed with 2.5 mL of distilled water and 0.5 mL of ferric chloride (0.1% *w*/*v*). The spectrophotometer was used to measure the absorbance of the resulting mixture at 700 nm. This procedure was repeated for the reference standards, gallic acid and butylated hydroxyl toluene. The percentage reducing power of the extracts was determined using the equation below:% Reducing power = A_0_ − A_s_/A_0_ × 100(3)
where A_0_ is the absorbance of the negative control and As is the absorbance of the extracts/standards.

## 5. Conclusions

The results of this study indicate that South African and Nigerian mistletoe tree ex- tracts are enriched with potent bioactive compounds and warrant further investigation via isolation and structural elucidation to find novel lead compounds, especially from the South African species, that may be developed and used for the treatment of various dis- eases. Over and above this, mistletoe samples from both countries are not very distinct chemically, judging from their TLC fingerprinting, which suggests the potential of using samples from both countries interchangeably. This study further indicated that Nigerian and South African mistletoe tree extracts contain similar phytochemicals as well as identical antioxidant activity. Thus, the West and South African plants are taxonomically well classified as African mistletoe trees because their chemistry is not significantly distinct.

## Figures and Tables

**Figure 1 plants-12-02016-f001:**
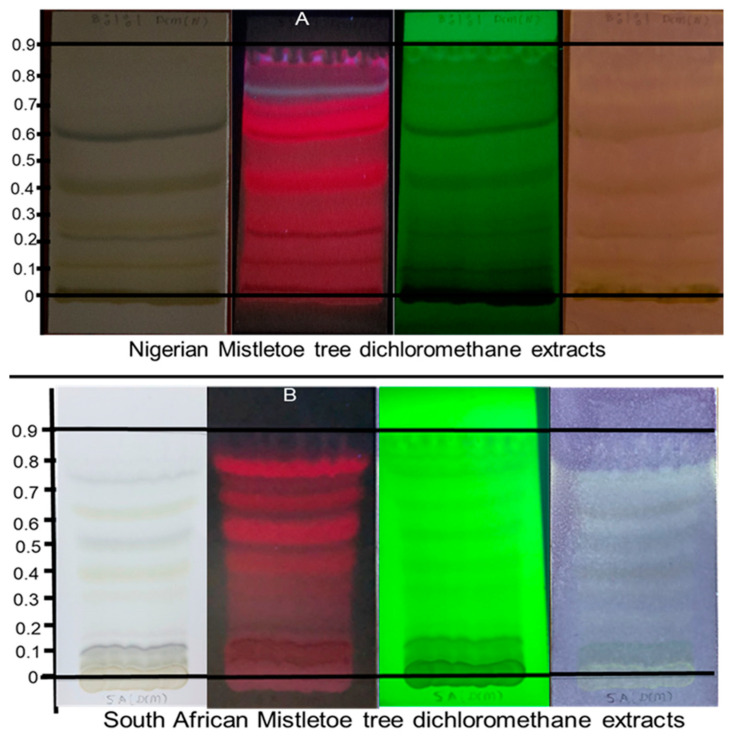
TLC chromatogram of NGDCM (**A**) and SADCM (**B**). Plates were viewed at visible light (plate 1), 366 nm (plate 2), 256 nm (plate 3), and sprayed with 0.2 mM DPPH/MeOH solution (plate 4).

**Figure 2 plants-12-02016-f002:**
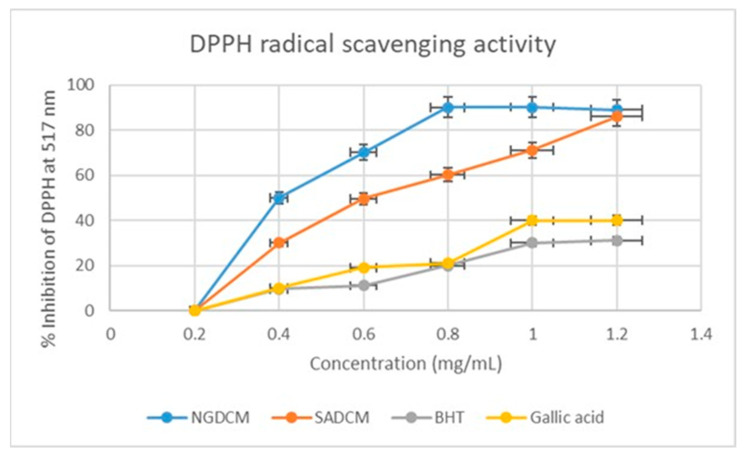
DPPH free radical scavenging percentage inhibition of NGDCM and SADCM mistletoe and standards. Each value is expressed as mean ± standard deviation of (*n* = 3).

**Figure 3 plants-12-02016-f003:**
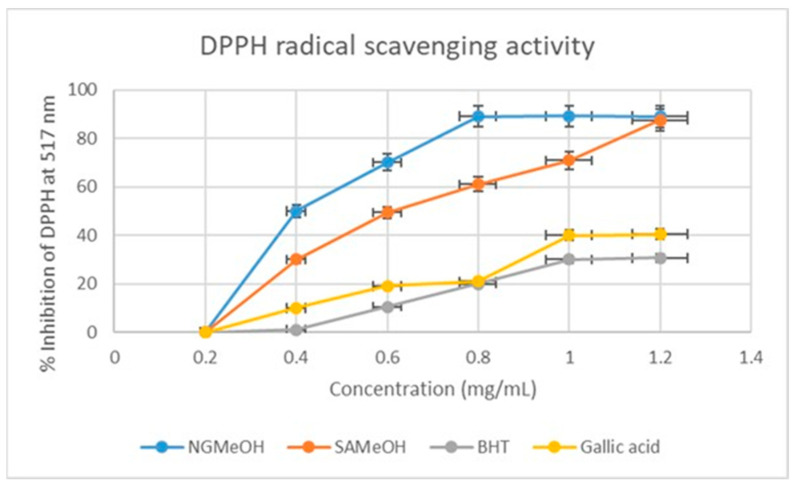
DPPH free radical scavenging percentage inhibition of the NGMeOH and SAMeOH mistletoe extracts and standards. Each value is expressed as mean ± standard deviation of (*n* = 3).

**Figure 4 plants-12-02016-f004:**
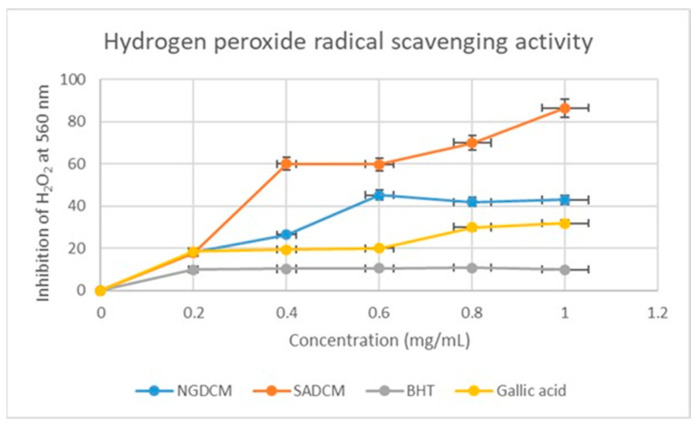
Hydrogen peroxide free radical scavenging percentage inhibition of NGDCM and SADCM mistletoe extracts and standards. Each value is expressed as mean ± standard deviation of (*n* = 3).

**Figure 5 plants-12-02016-f005:**
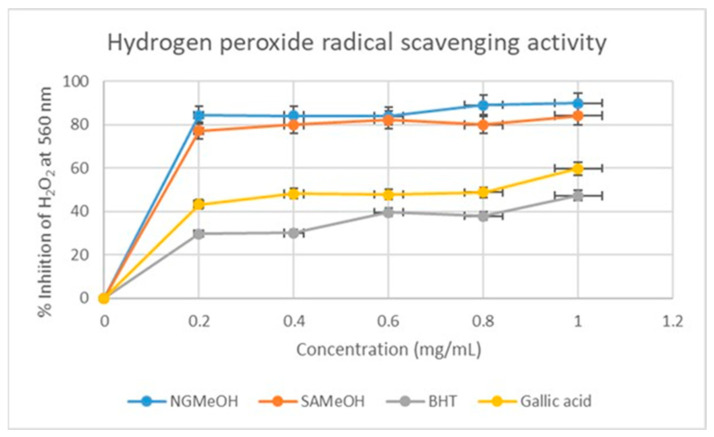
Hydrogen peroxide free radical scavenging percentage inhibition of NGMeOH and SAMeOH mistletoe extracts and standards. Each value is expressed as mean ± standard deviation of (*n* = 3).

**Figure 6 plants-12-02016-f006:**
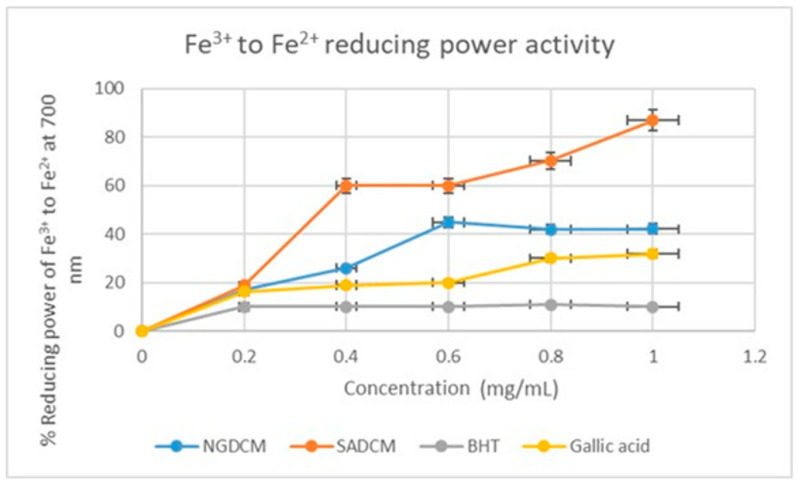
Ferric chloride reducing power activity of NGDCM and SADCM mistletoe extracts and standards. Each value is expressed as mean ± standard deviation of (*n* = 3).

**Figure 7 plants-12-02016-f007:**
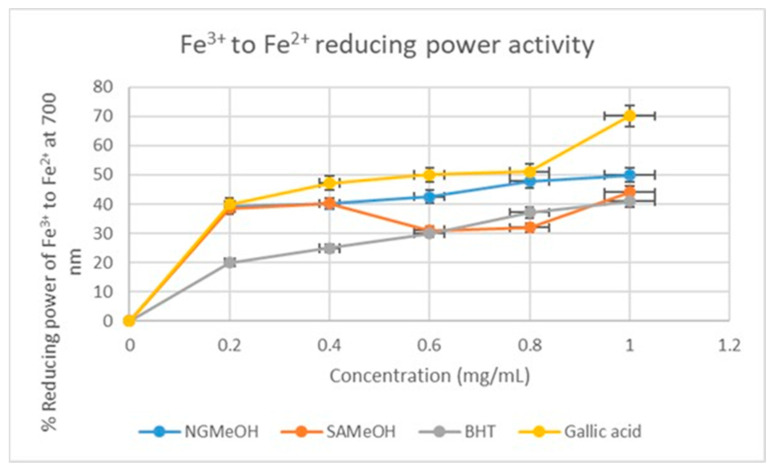
Ferric chloride reducing power activity of NGMeOH and SAMeOH mistletoe extracts and standards. Each value is expressed as mean ± standard deviation of (*n* = 3).

**Table 1 plants-12-02016-t001:** Qualitative phytochemical analysis indicating the presence of different phytochemicals in South African and Nigerian mistletoe extracts.

Phytochemicals	Mistletoe Extracts Investigated			
	NGDCM	SADCM	NGMeOH	SAMeOH
1. Alkaloids	−	−	−	−
2. Flavonoids	−	−	++	++
3. Cardiac glycosides	++	+	++	+
4. Saponins	++	++	+++	++++
5. Phenolics	+	+	+++	+++
6. Phlobatannins	−	−	++	++
7. Tannins	++	++	+++	++++
8. Terpenoids	+	+	++	++
9. Anthraquinones	−	−	−	−

Absent (−), Trace amount present (+), Moderate amount present (++) and strongly present (+++).

**Table 2 plants-12-02016-t002:** Phytochemical bands and their antioxidant potentials for SADCM and NGDCM MTT extracts.

Mistletoe Tree Extracts				
Band Number	SADCM		NGDCM	
	Rf Value	Antioxidant	Rf Value	Antioxidant
1	0	++	0	++
2	0.1	−	0.12	−
3	0.15	−	0.22	+
4	0.3	++	0.27	−
5	0.47	++	0.42	−
6	0.53	++	0.62	−
7	0.6	++	0.83	++
8	0.73	+++	0.85	++

Absent (−), Trace amount present (+), Moderate amount present (++) and strongly present (+++).

**Table 3 plants-12-02016-t003:** Phytochemical bands and their antioxidant potentials for SAMeOH and NGMeOH mistletoe.

	Mistletoe Extracts Investigated			
Band Number	NGMeOH		SAMeOH	
	Rf Value	Antioxidant	Rf Value	Antioxidant
1	0	+++	0	−
2	0.1	+++	0.10	−
3	0.31	+++	0.20	−
4	0.43	+++	0.30	−
5	0.53	+	0.40	−
6	0.61	−	0.52	−
7	0.7	−	0.62	−
8	0.75	−	0.73	−
9	0.8	−	0.75	−

Absent (−), Trace amount present (+), Moderate amount present (++) and strongly present (+++)

**Table 4 plants-12-02016-t004:** IC_50_ values indicating the potential inhibition by the four mistletoe extracts.

Extracts and Standards		IC50 mg/mL	
	DPPH Scavenging	H_2_O_2_ Scavenging	Reducing Power
NGDCM	0.51	0.95	0.60
SADCM	0.31	0.51	0.76
NGMeOH	0.20	0.17	1.15
SAMeOH	0.51	0.23	0.86
Gallic acid	1.17	1.54	0.62
BHT	1.47	1.04	1.15

**Table 5 plants-12-02016-t005:** Standard protocols used for qualitative phytochemical screening.

Phytochemical Group	Qualitative Chemical Test Protocol	Reference
*Alkaloids-Anticho- linesterase*	A total of 5.0 % of aqueous HCl + about 0.5 g of plant extract. Filter and treat 1 mL of filtrate with a few drops of Dragendorff reagent. The presence of precipitates in the mixture would indicate a preliminary positive test for alkaloids. A total of 0.5 g plant extract + a few drops of magnesium strip + drops of concentrated H_2_SO_4_. A red coloration indicates a positive test for flavonoids. A total of 0.5 g of dry powdered material is boiled in 20 mL of water and filtered. A few drops of 0.1 % ferric chloride are added to the filtrate. A brownish, green to blue-black coloration indicates a positive test for tannins. The aqueous extract of the plant is boiled in 1.0% aqueous HCl acid. A positive test for phlobatannins is indicated by deposition of red precipitate. Say 2.0 g of powdered plant material is boiled in 20 mL of distilled water and then filtered. A total of 10 mL of the filtrate is further mixed with 5.0 mL of distilled water and shaken vigorously till a stable froth is formed. The froth is finally mix with a few drops of olive oil and shaken vigorously. The formation of an emulsion indicates a positive test for saponins. About 2.0 mL of acetic anhydride is added to 0.5 g ethanolic extract of the plant material. To this mix should be added 2.0 mL of H_2_SO_4_. The color of the mixture changing from violet to blue or green indicates a positive test for steroids. Mix 5.0 mL of plant extract with 2.0 mL of CHCl_3_, and to the mixture add 3.0 mL of concentrated H_2_SO_4_ acid in drops to form a layer. A red-to-brown coloration on the inter-phase of the mixture is a positive result for terpenoids. A total of 5.0 mL of the plant extract is usually treated with 2.0 mL of glacial acetic acid containing a few drops of ferric chloride solution. This should be treated with 1 mL of concentrate of H_2_SO_4_. If a brown ring is formed at the interphase of the solution, a deoxy sugar of cardenolides is present. A violet ring may also form below the brown ring. The Borntrager’s test is used to test for this class of plant metabolites. It involves mixing about 5.0 g of plant extract with 10 mL of benzene, shaking vigorously and then filtering. To the filtrate should be added 10 % ammonia solution. The entire mixture is shaken. The presence of a pink, red, or violet color in the lower phase of the mixtures is a positive test for hydroxyl anthraquinones.	[32]
*Flavonoids*		[33]
*Tannins*		[34]
*Phlobatannins*		[35]
*Saponins*		[35]
*Steroids*		[34]
*Terpenoids*		[36]
*Cardiac glycosides*		[37]
*Free anthraquinones*		[34]

## Data Availability

The data used in the current study is contained within the article.

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
