# Peer review of "Phytochemical Profiling, and Antioxidant Potentials of South African and Nigerian Loranthus micranthus Linn.: The African Mistletoe Exposé"

_plants, 2023, doi:10.3390/plants12102016_

Round 1
Reviewer 1 Report
The current work is not suitable for publication with respect to high-impacted specialized Journals such as Plants.
Major concern
The reported work in this MS is lacking novelty, the qualitative determination of plant constituents was done using preliminary screening chemical tests. Standardization of both plant extracts should be carried out using LCMS. A comparison of the metabolic profiles of both plant extracts should be carried out. We cannot depend mainly in these preliminary phytochemical screening tests, because these tests may give positive results with other plant constituents. Therefore, metabolomics and LCMS evaluations are important to increase the value of the current work.
The MS is not well written, the results section seems to be a discussion section. They should be merged under on section results and discussion.
What are the reported constituents from this plant?. What are the reported biological activities and folk uses in both country? This should be carefully discussed.
Minor concerns
1- The plant genus and species names should be italicized through the whole MS.
2- Line 8-12. Should be summarize, the abstract should be concise and more informative.
3- Abstracts needs careful revision. The control results should be added.
4- English needs careful revision. There are many typing and grammatical mistakes throughout the whole MS.
5- Full names for all abbreviations should be added when they are first appeared.
6- Plant family name should be added in the abstract and keywords.
Author Response
REVIEWER 1 SUGGESTIONS |
AUTHOR RESPONCE |
LINE NUMBER |
Extensive editing of English language and style required |
|
|
Does the introduction provide sufficient background and include all relevant references? (x) Must be improved |
The introduction has been improved and relevant references also included |
L47, L56 – L65 |
Are all the cited references relevant to the research? (x) Must be improved |
Has been improved and yes, all cited references are relevant to the study |
|
Is the research design appropriate? (x) Must be improved |
The research designed has been improved |
L382- 391, L397 – 403, L411 - 420 |
Are the methods adequately described? (x) Must be improved |
|
|
Are the results clearly presented? (x) Must be improved |
|
|
Are the conclusions supported by the results? (x) Must be improved |
The conclusions had been improved |
L394 - 401 |
The reported work in this MS is lacking novelty, the qualitative determination of plant constituents was done using preliminary screening chemical tests. Standardization of both plant extracts should be carried out using LCMS |
The aim of the study is not to standardize the plant extracts but to investigate of the Nigerian and the South species have the same phytochemical profiles using standard protocols |
Table 1 (L130, L and Table 5 (L365) |
A comparison of the metabolic profiles of both plant extracts should be carried out |
|
|
We cannot depend mainly in these preliminary phytochemical screening tests because these tests may give positive results with other plant constituents. Therefore, metabolomics and LCMS evaluations are important to increase the value of the current work. |
The study aim was not profiling the plants’ metabolome which requires LC-MS.
In addition to phytochemical profiling, the other objective of the study was to investigate the antioxidant potentials of SA and NG mistletoe extracts suing DPPH, H2O2 and Fe3+ to Fe2+ assays |
Table 2 (L160,), Fig 2 (L171), Table 3 (L181), Figs 3 – 7, Table 4 (L275), section 4.4 – 4.5. |
The MS is not well written, the results section seems to be a discussion section. They should be merged under on section results and discussion. |
The instruction for authors prescribes the results and discussion section to be reported as sections 2 and 3 respectively. We have done so in the manuscript. |
L66 and L252 |
What are the reported constituents from this plant? What are the reported biological activities and folk uses in both country? This should be carefully discussed. |
Reported compounds from the plant, the folkloric uses and biological activities of the plant has been included |
L285 - 296 |
Minor concerns |
|
|
1- The plant genus and species names should be italicized through the whole MS. |
Plant genus and species has been italicized throughout the document. |
|
2- Line 8-12. Should be summarize, the abstract should be concise and more informative. |
Line 8 -12 has been summarized with fewer phrases. |
Line 8 - 11 |
3- Abstracts needs careful revision. The control results should be added. |
Abstract has been revised and results for the control -gallic acid and BHT included. |
L24 - 31 |
4- English needs careful revision. There are many typing and grammatical mistakes throughout the whole MS. |
English language has been carefully revised |
L1 - 519 |
5- Full names for all abbreviations should be added when they are first appeared. |
Done |
L14, 16, 19, 20, 28, 186 and 267 |
6- Plant family name should be added in the abstract and keywords. |
Plant family name ‘Linn.’ has been added to the abstract and keywords |
L 8, L9, 13 and L35 |

Reviewer 2 Report
Similar work was described in (2014) Phytochemical Analysis of Loranthus micranthus Extracts and Their in vitro Antioxidant and Antibacterial Activities, Journal of Biologically Active Products from Nature, 4:4, 303-315, DOI: 10.1080/22311866.2014.936901
I could not detect any novelty in the present study. Similar methods wore used as the above-cited reference. Besides, the reference has not been cited and commented on in the present study.
The phytochemical procedures were highly rudimental and included some chemical reactions and thin-layer chromatography.
The authors should try using LC-MS in order to tentatively identify the particular chemical constituents responsible for the antioxidant activity.
In my opinion, the manuscript should be rejected.
Author Response
Please see the responses in the attachment.

Reviewer 3 Report
This manuscrips wants to determine the differences between the mistletoes coming from South Africa and Nigeria.
The introduction should be completed with more convenient references adapted to the different topics treated and make a state of the art about the already known knowledge on mistletoes.
The results section is globaly ok eventhough it is necessary to make some statistical analysis before stating things on the comparisons. Need also in the figures and tables.
The discussion section should be completed reviewed to make a real discussion on the results obtained and not doing a summary of the results again.
The material and methods section is globally ok.
You could find more detailed comments in the document attached

Author Response
REVIEWER 2 SUGGESTIONS |
AUTHOR RESPONCE |
LINE NUMBER |
I am not qualified to assess the quality of English in this paper |
Thank you |
Not applicable (NA) |
Does the introduction provide sufficient background and include all relevant references? (x) Must be improved) |
The introduction has been improved and relevant references also included |
L47, L56 – L65 |
Are all the cited references relevant to the research? (x) Must be improved) |
Yes, all references are relevant to the research Please search the reference list |
L58, 59, 66, 70, 306, 437 to 517 |
Is the research design appropriate? (x) Not applicable |
NA |
NA |
Are the methods adequately described? (x) Yes |
The methods have been improved adequately described |
|
Are the results clearly presented? (x) Must be improved |
The results have been improved |
L181 – 182, L266 – 271, L277 - 278 |
Are the conclusions supported by the results? (x) Must be improved |
The conclusions had been improved |
L394 - 401 |
The introduction should be completed with more convenient references adapted to the different topics treated and make a state of the art about the already known knowledge on mistletoes. |
Done |
L48, 49, L57 – L66 |
The results section is globally ok even though it is necessary to make some statistical analysis before stating things on the comparisons. Need also in the figures and tables. |
A p = 0.06 was calculated for Table 2 A p = 0.002 has been determined for Table 3 p-vales for the IC50 values of the tree antioxidant assay methods has determined and discussed. |
L158 – 159 L181 – 182 L266 - 271 |
The discussion section should be completed reviewed to make a real discussion on the results obtained and not doing a summary of the results again. |
The discussion section has been improved. |
L312 - 324 |
The material and methods section are globally ok. |
Thank you |
|
You could find more detailed comments in the document attached. |
Detailed comments seen and attended to |
|

Round 2
Reviewer 1 Report
No comments
Author Response
Thank you for your time.
Reviewer 3 Report
The introduction has been increased of quality with better convenient references. But the discussion is still lacking of references, and a real discussion of the results.
But my main problem concerns the figures in the results section, I'm worried about the fact that standard deviation you say you have calculated never appear in the figures and considering the methods you used the level of precision can not be so high to give standard errors which are hidden by the dots all the time. So I cannot evaluate the "significance" of the results you are proposing.
In addition you have other minor remarks in the file attached.

Author Response
REVIEWER 2 SUGGESTIONS 2 |
AUTHOR RESPONCE |
LINE NUMBER |
The introduction has been increased of quality with better convenient references. But the discussion is still lacking of references, and a real discussion of the results. |
They are three references [29], [30] and [31] in the discussions |
L278, l294, and L313 |
But my main problem concerns the figures in the results section, I'm worried about the fact that standard deviation you say you have calculated never appear in the figures and considering the methods you used the level of precision can not be so high to give standard errors which are hidden by the dots all the time. So I cannot evaluate the "significance" of the results you are proposing. |
If the results were presented in a tabular form, then standard deviations would have been included. Standard deviations cant be represented in a figure(s).
Indication of error bars does not feature in all graphs. |
|
In addition you have other minor remarks in the file attached |
||
Present |
Corrected to presence |
|
Table 4: But the signs are not in the Table |
-, +, ++, +++ legend has been removed from Table 4 |
L89 |
P |
Corrected to p |
L111 |
P |
Corrected to p |
L112 |
P |
Corrected to p |
L113 |
Tannins |
Corrected to tannins |
L117 |
Figure 1, indicate in the figure the number of plates |
Numbers 1 – 4 added to the plates |
L151 |
366nm, add space |
Space added |
L152 |
has |
Changed to had |
L156 |
Indicate number also in photo |
Numbers indicated |
L173 |
was |
Changed to were |
L189 |
Figure 3, unit missing |
Unit (nm) added |
Figure |
NGDCM The legend should be explicit Anywhere I can see standard deviation |
Meaning of NGDCM was given
Standard cant be shown on a Figure but in a table |
L20
Figure 3 and not a Table |
Unit missing on Figure |
Unit are affixed on all figures |
Figures 3 - 8 |
0,2 |
Change to 0.2 |
L220 |
Figure 4; is really extrange that we cannot see the error bars in any point in any figure. |
Error bars are not mandatory to all graphs |
L210 and L211 |
use |
Changed to used |
L235 |
South African and Nigerian DCM…Add acronym |
SADCM and NGDCM added |
L252 |
Repeated information |
Deleted |
L310 |

Round 3
Reviewer 3 Report
Only minor corrections have been done.
But nothing have been done to correct the statistics associated to each figure like adding the error bars.
Author Response
Please see an updated summary of responses in the attachment.

Round 4
Reviewer 3 Report
The manuscript has been improved, specially the statistical part.
Now only minor corrections have to been performed.
You have in the attached document, some correction along the document.
Other than the form and some sentences clarification, in the mat & method section a number is missing.

Author Response
Asl?? |
Asl has been deleted |
L62 |
=0.002 |
Corrected p = 0.002, 180 typo also removed |
L179 |
Strike through |
Deleted |
L281 |
Strike through |
Deleted |
L382 |
- Was present |
Deleted |
L350 |
In particular, mistletoe is in the |
Corrected to Mistletoe is used in the traditional management |
290 |
Standards protocols |
Corrected to standard protocols |
L344 |
… in the mat & method section a number is missing. |
Missing number not detected |
Not applicable |
|
|
|